# HD-OCT Angiography and SD-OCT in Patients with Mild or No Clinically Apparent Diabetic Retinopathy

**DOI:** 10.3390/biomedicines13051251

**Published:** 2025-05-20

**Authors:** Maja Vinković, Andrijana Kopić, Tvrtka Benašić, Dubravka Biuk, Ivanka Maduna, Stela Vujosevic

**Affiliations:** 1Department of Ophthalmology, University Hospital Centre Osijek, 31000 Osijek, Croatia; andrijanakopic@gmail.com (A.K.); tvrtka@gmail.com (T.B.); dubravka.biuk@gmail.com (D.B.); 2Faculty of Medicine Osijek, Josip Juraj Strossmayer University of Osijek, 31000 Osijek, Croatia; ivankamadun@gmail.com; 3Health Centre of Osijek-Baranja County, 31000 Osijek, Croatia; 4Department of Biomedical, Surgical and Dental Sciences, University of Milan, 20122 Milan, Italy; stela.vujosevic@gmail.com; 5Eye Clinic, IRCCS MultiMedica, 20138 Milan, Italy

**Keywords:** OCT angiography, diabetic retinopathy, microvascular density, retinal ganglion cells, tomography, optical coherence

## Abstract

**Purpose:** To analyze the retinal and choriocapillaris changes in diabetic patients with no or with early signs of diabetic retinopathy using high-definition (HD) angio optical coherence tomography angiography (OCTA) software and spectral-domain (SD) OCT. **Methods:** A total of 112 eyes (54 eyes from 27 diabetic patients and 58 eyes from 29 control subjects) were included in this retrospective cross-sectional study of healthy and diabetic adults. Retinal microvascular changes were assessed by using HD-OCTA software to calculate vascular density (VD) and foveal avascular zone (FAZ). SD-OCT was used to assess retinal thickness and volume in parafovea as well as ganglion cell complex (GCC) parameters. **Results:** The VD-whole image was significantly higher in the healthy control group (MW z = 1109.5, *p* = 0.012; *t* = 2.611, *p* = 0.010). Also, VD-parafovea was significantly higher in the healthy subjects (MW z = 1053.5, *p* = 0.004; *t* = 3.207, *p* = 0.002). GCC focal loss volume (FLV) was significantly decreased in diabetic patients (*p* = 0.051). Non-flow FAZ did not show a statistically significant difference between groups, although the FAZ was larger in the diabetic patients. **Conclusions:** Diabetic patients with no or early signs of diabetic retinopathy have decreased VD compared to healthy individuals. They also present retinal changes at the GCC that are correlated with initial neurodegeneration. HD-OCTA and SD-OCT can detect vascular changes and structural signs of retinal neurodegeneration before clinically apparent diabetic retinopathy. Potentially, these methods may offer new biomarkers for monitoring disease progression and visual prognosis.

## 1. Introduction

Diabetic retinopathy (DR) is a major cause of visual impairment and blindness among the working-age population [1]. Although ophthalmoscopy and fundus photography are the standard methods for diagnosing and staging DR, they lack the sensitivity to detect early changes, which are crucial for preventing disease progression. Spectral-domain (SD) optical coherence tomography angiography (OCTA) is a diagnostic technique that noninvasively, safely and quickly images retinal microvasculature with better resolution than fluorescein angiography (FFA) [2,3]. It allows for qualitative (impaired capillary perfusion, intraretinal microvascular abnormalities, neovascularization and intraretinal fluid) and quantitative assessment [4]. FFA is considered the standard technique that establishes quantitative indices of perfusion in DR [5]. However, it does not detect subtle changes, especially in the deep capillary plexus [6]. OCTA has been used to determine retinal perfusion in normal subjects and retinal vascular disease [7,8,9,10]. The OCTA evaluation in diabetic patients revealed an enlargement of the vascular arcades of the foveal avascular zone (FAZ) and notable reductions in reduced capillary density predominantly in the deep layer [11,12]. Retinal neurodegeneration can be found in the early stages of diabetes, preceding the appearance of clinically detectable microvascular changes [13].

In this study, we used an SD-OCTA device to image retinal changes in diabetic subjects with early DR or no clinically apparent DR and compared them with a control group of healthy subjects.

Furthermore, we aimed to study correlations between vascular density parameters and structural parameters such as GCC thickness, as well as duration of diabetes. 

## 2. Patients and Methods

A total of 56 patients (112 eyes) were included in the study. Patients were divided into two groups: healthy subjects (n = 29) and patients with type 2 diabetes mellitus (n = 27). All patients underwent a clinical examination, including OCTA imaging, at the Eye Clinic University Hospital Osijek from September 2018 to June 2019.

The study was conducted following the principles of the Helsinki Declaration and approved by the institutional review board. Informed consent was obtained for each subject. All participants had a complete eye examination, which included BCVA, slit-lamp and fundus examination. A complete medical history was taken for every patient, including HbA1c level and type and duration of diabetes. Both eyes were examined by SD-OCT after pupillary dilation.

The inclusion criteria were adult diabetic patients who had no clinically apparent DR or mild DR without DME, HbA1c ≤ 9% and BCVA logMAR 0.

The exclusion criteria were prior intraocular surgery, glaucoma, ocular hypertension, other retinal vascular disorders, astygmatismus ≥ 3 Diopter (dpt), spherical refractive error ≥ 6 dpt and optical media opacities that interfere with visualization of the fundus.

The signal stress index of 40 or more was selected as a cut-off point on the OCT device.

Retinal morphology data were obtained by using the AngioVue OCTA system (RTVue-XR Avanti; Optovue, Fremont, CA, USA) with an SSADA (split-spectrum amplitude-decorrelation angiography) software algorithm (v2014.2.0.90). We used GCC scan (6 × 6), retina thickness map (5 × 5) and HD angio (6 × 6) protocol. The following parameters were evaluated: full retinal thickness and volume. Retinal tissue layers’ thickness data were expressed as mean values using retina map software. Retinal ganglion cell complex (GCC) thickness (which consists of the retinal nerve fiber layer, ganglion cell layer and the inner plexiform layer) values were analyzed and compared to age-matched normative database. The GCC thickness was defined as the distance from the internal limiting membrane to the boundary of the outer inner plexiform layer. In addition to the calculated mean thicknesses, two pattern-based diagnostic parameters were used: focal loss volume (FLV) and global loss volume (GLV). FLV and GLV are two parameters that measure the amount of significant GCC loss. FLV measures the average amount of focal loss over the entire GCC map, whereas GLV measures the average amount of GCC loss. Also, we evaluated vascular density (VD) in the whole image, VD in the fovea and VD in the parafovea. The foveal avascular zone (FAZ) area (mm^2^) was evaluated by using the non-flow area tool of the software that delineated it automatically after selecting a segment of the FAZ (Figure 1). The superficial retinal plexus (SCP), deep retinal plexus (DCP) and choriocapillary vascular network (CC) were obtained by using an automated software algorithm. The boundaries, automatically detected by the instrument for each layer, were as follows: a slab extending from 3 to 15 μm from the internal limiting membrane was generated for detecting the SCP, a slab extending from 15 to 70 μm below the internal limiting membrane for the DCP, and a slab extending from 30 to 60 μm below retinal pigment epithelium reference for CC. 

### Statistics

An unpaired *t*-test was used to compare retinal layers’ structure and OCTA parameters between control subjects and patients with diabetes mellitus (DM). The Mann–Whitney U test was used for nonparametric data. Pearson’s coefficient of correlation was used to check the correlation between the duration of diabetes, retinal structure (total GCC) and flow parameters (VD, FAZ).

The analysis of the scans obtained with the OCTA and demographic data was performed with IBM SPSS Statistics Version 25.0, Armonk, NY, USA: IBM Corp. A *p* value < 0.05 was considered statistically significant. We performed multivariate linear regression analysis and Hierarchical Linear Regression in JASP Team (2024). JASP version 0.19.3.

## 3. Results

Among the total of 56 patients, there were 22 male and 34 female; the mean age was 51.4 years. Demographics, as well as duration of diabetes and the level of HbA1c, are reported in Table 1. In the diabetic group, the HbA1c mean was 7.4% ± 1.27, and the mean duration of diabetes was 6 years ± 2 years.

The difference in the mean age between healthy subjects and the diabetic group of patients is tested with a *t*-test for independent samples. There is no difference in age between groups (*p* = 0.069); however, there were more male subjects in the diabetes group than in the control group (55.6% vs. 44.1%).

Retinal thickness in the macular and parafoveal regions did not differ significantly between the two groups (*t* = 1.6, *p* = 0.113). The difference between results in the two groups of patients was confirmed for the following variables: VD-whole image (%) and VD-parafovea and VD-perifovea (*t*-test for independent samples). Descriptive data for parameters in OCTA are shown in Table 2. The VD-whole image was significantly higher in the healthy control group (MW z = 1109.5, *p* = 0.012; *t* = 2.611, *p* = 0.010) and VD-parafovea was significantly higher in the healthy subjects (MW z = 1053.5, *p* = 0.004; *t* = 3.207, *p* = 0.002). The difference between results in the two groups of patients was confirmed for the following variables: outer retinal flow, VD-whole image SCP and DCP, VD-parafovea SCP and DCP, and VD-perifovea SCP and DCP (*p* = 0.004, *p* < 0.001, *p* = 0.012, *p* < 0.001, *p* <0.001, *p* < 0.01, *p* = 0.027, respectively).

Non-flow FAZ did not show a statistically significant difference between groups, although the FAZ was larger in the diabetic patients (*p* = 0.189).

A multiple linear regression was performed to examine the effect of age and gender on VD. Models shown in Table 3, except SCP VD fovea and DCP VD fovea models, were statistically significant. The initial model included the age and gender of all participants as predictors of VD. Age was a significant negative predictor in all initial models, indicating that VD decreased with age. Gender was also significant in SCP VD models, with females showing higher VD compared to males. No collinearity issues were detected. A second model was tested to determine whether the relationship between age and vessel density differed by group (people with diabetes vs. controls) by including an interaction term (Age × Group). The interaction was not statistically significant, suggesting that the effect of age on VD did not differ significantly between groups. A hierarchical linear regression (Table 4) was conducted to examine predictors of SCP VD in the whole image in diabetic eyes. In the first step, age and gender were entered, explaining 31% of the variance (*p* = 0.096). In step two, adding HbA1c increased the model’s explanatory power to 40.8%, with HbA1c showing a borderline association with VD (B = −0.278, *p* = 0.053). In the final model, the inclusion of diabetes duration significantly improved the model (ΔR^2^ = 0.288, *p* = 0.003), raising the explained variance to 53.7%. Duration (B = −0.387, *p* = 0.007) and age (B = −0.313, *p* = 0.020) were both significant negative predictors of VD, indicating that older age and longer disease duration are associated with lower retinal VD in diabetic patients.

Total GCC thickness was decreased in diabetic patients, but the difference was not significant (*p* = 0.298). We found a significant focal thinning of GCC in the diabetic group of patients (*p* = 0.051). It has no preference for either the superior or inferior halves of the macula (Table 5).

We found a weak positive correlation between the duration of diabetes and FAZ area and also a weak inverse correlation between the duration of diabetes with VD-whole image and GCC total (Table 6).

## 4. Discussion

Recently, a number of studies have presented both qualitative and quantitative OCTA findings in diabetic patients [2,4,14]. Kim et al. have reported progressively decreasing capillary density, branching complexity and progressively increasing average vascular caliber across different DR stages [15]. They have not been able to detect a significant difference in these variables between healthy subjects and patients with mild non-proliferative DR. This discrepancy from the present study may be caused by differences in equipment and/or smaller control groups in their study. Significantly reduced VD in the SCP and DCP in mild non-proliferative DR compared to healthy controls has also been observed in the study of Agemy et al. [7]. Previous studies have reported FAZ enlargement in diabetic patients without DR [16,17,18]. In the present study, the FAZ area of diabetic patients with no or mild DR was also increased in comparison to controls, but the difference failed to prove significant (*p* = 0.183). We believe this would be significant for the larger number of patients. FAZ enlargement was consistent with the reduced VD in both SCP and DCP observed in diabetic patients with no or mild DR in this study. These findings suggest compromised circulation in the inner retinal layers before manifested DR.

This study suggests that both age and gender are associated with VD, with VD in both SCP and DCP decreasing with age, while diabetic disease does not significantly change the effect of age on VD. You et al. [19] reported significantly lower VD in healthy eyes in the male gender, which was also reported in our study, while the effect of age was borderline (*p* = 0.09). Shahlaee et al. observed also noted negative correlations with age [10].

It has also been suggested that ischemia at the DCP may play an important role in the changes in the outer retina detected with SD-OCT [20,21]. A study by Ong et al. [22] showed that changes in DCP in treatment-naïve moderate to severe NPDR can predict significant clinical DR complications at 1 year. A recent study indicated that DCP nonperfusion has high specificity and sensitivity for identifying eyes with clinically referable DR [23]. Our study showed that changes in parafovea and perifovea in the DCP are present even in diabetic patients with no or mild DR. There is a need for prospective longitudinal studies evaluating early OCTA diabetic changes in the prediction of significant long-term clinical DR outcomes and early personalized diabetes regulation. As this study is cross-sectional, we are not able to conclude whether changes in perifovea come before changes in parafovea, potentially indicating early DR biomarkers. One prospective longitudinal study on diabetic patients’ eyes ranging from mild NPDR to PDR over one year found that a larger decrease in SCP VD was associated with worsening retinal sensitivity [24]. Our study showed that changes in SCP VD whole image, perifovea and parafovea are present at the beginning of DR, even before clinical DR and while patients are typically asymptomatic. In our study, the duration of diabetes explained an additional 28.8% of the variance in Model 3 (including age, gender and HBA1c value), showing a significant influence on SCD VD in the whole image. A study by Li et al. also reported a negative correlation between diabetes duration on SCP and DCP VD, while no significant correlation was found with HbA1c [25].

Growing evidence indicates that the pathogenesis of early DR involves both neurodegenerations, including retinal ganglion cell loss (RGCs) and microvascular changes. [26,27]. GCC thinning is associated with an increased cup-to-disc ratio in diabetic eyes [28]. Significant GCC thinning was found in our group of diabetic patients, which is mainly focal rather than diffuse and shows no predilection for the superior or inferior macular regions. These findings were also confirmed in a study by Hegazy et al. [29].

We found a weak positive correlation between the duration of diabetes and FAZ area (mm^2^) and also a weak inverse correlation between the duration of diabetes with VD-whole image and GCC total. These results should be evaluated in further longitudinal studies.

The RTVue-100 has a 5 µm axial resolution and a scan rate of 26,000 axial scans/s. In addition, the RTVue-100 has software embedded that can provide five macular maps (MM5), the thickness of the RNFL and the thickness of the ganglion cell complex (GCC) [27,30,31]. Thus, the RTVue-100 can be useful for determining whether eyes at the early stage of DR have distinct pathognomonic features of early-stage DR.

There were no significant differences in retinal thickness between controls and diabetic patients, suggesting that microvascular changes may occur before structural retinal alterations. However, longitudinal studies are necessary to confirm this relationship and better understand the progression of diabetic retinal changes [20].

It has also been suggested that ischemia at the DCP may play an important role in the changes in the outer retina detected with SD-OCT [21].

Previous studies showed that choroidal circulation, assessed through color Doppler imaging of posterior ciliary arteries, is significantly decreased in patients with DR [32]. Similarly, our study observed a trend toward lower choriocapillaris VD in diabetic individuals compared to controls. This may suggest that impairments in both retinal and choroidal circulation may occur prior to the clinical onset of DR, indicating a potential interconnection in the vascular changes in both systems.

We believe that the presence of concomitant changes in GCC and vascular alterations detected as decreased density and enlargement of FAZ indicate simultaneous changes in the neurovascular unit and further support the neuro-vascular pathogenesis of DR. In conclusion, results from this study suggest that SCP and DCP VD in the parafovea of diabetic patients without DR is decreased as compared to healthy subjects.

Automated quantitative algorithms allow for objective assessment of retinal vascular and structural changes in eyes with mild or no DR, which has also been confirmed by other authors. This cross-sectional study has several limitations. Due to its design, it can only demonstrate associations between diabetes and changes in VD without establishing causality. The sample size was relatively small, and due to the small sample size, we included both eyes of the subjects, which may limit the generalizability and statistical power of the findings. Additionally, potential confounding factors such as glycemic control, blood pressure and other systemic conditions were not accounted for, which could influence vascular measurements. Lastly, measurement variability or limitations in imaging resolution and analysis software may have affected the accuracy of VD assessment.

The results also suggest that OCT angiography parameters may be potential biomarkers for detecting diabetic eyes at risk of developing retinopathy. Future studies on a larger number of patients are needed to confirm these results.

## Figures and Tables

**Figure 1 biomedicines-13-01251-f001:**
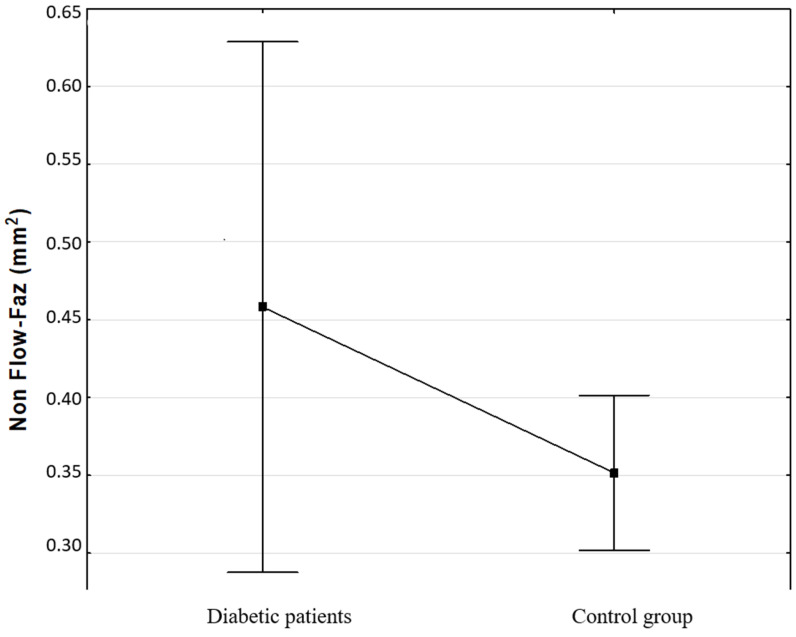
Arithmetic mean non-flow FAZ of control and diabetic group of patients with 95% CI.

**Table 1 biomedicines-13-01251-t001:** Demographic data.

	Diabetic Patients	Control
Patients (N)	27	29
Eyes (N)	54	58
Gender (N)	M = 15; F = 12	M = 12; F = 17
Age (years) *	52.3 ± 10.8	49.0 ± 10.3
Duration of DM (months) *	83.4 ± 24.3	-
HBA1c (%) *	7.3 ± 1.2	-

* The data presented are mean ± standard deviation. DM, diabetes mellitus; M, male; F, female; HBA1c, hemoglobin A1c.

**Table 2 biomedicines-13-01251-t002:** Angio parameters according to group of patients—descriptive data.

	Group	Mean	Standard Deviation	Difference	t	*p **
Flow Outer Retinal (mm^2^)	Diabetic patients (N = 49)	1.19	0.17	−0.11	−2.914	0.004
	Control (N = 48)	1.08	0.20			
Flow Choroid Capillaris (mm^2^)	Diabetic patients (N = 48)	1.87	0.082	0.028	1.807	0.07
	Control (N = 48)	1.9	0.07			
VD SCP—whole image (%)	Diabetic patients (N = 51)	45	4.53	3.31	3.702	<0.001
	Control (N = 47)	48.31	4.29			
VD DCP—whole image (%)	Diabetic patients (N = 50)	45	6.06	3.11	2.56	0.012
	Control (N = 47)	48.11	5.92			
VD SCP—fovea (%)	Diabetic patients (N = 51)	18.89	7.53	0.12	0.0823	0.963
	Control (N = 47)	19.02	7.23			
VD DCP—fovea (%)	Diabetic patients (N = 50)	34.04	8.82	1.74	1.006	0.317
	Control (N = 47)	35.78	8.13			
VD SCP—parafovea (%)	Diabetic patients (N = 51)	45.10	6.89	4.45	3.61	<0.001
	Control (N = 47)	49.55	5.03			
VD DCP—parafovea (%)	Diabetic patients (N = 50)	50.84	4.77	3.39	3.57	<0.001
	Control (N = 47)	54.23	4.56			
VD SCP—perifovea (%)	Diabetic patients (N = 51)	46.13	4.25	3.34	3.75	<0.001
	Control (N = 46)	49.47	4.52			
VD DCP—perifovea (%)	Diabetic patients (N = 50)	45.96	6.67	3.07	2.253	0.027
	Control (N = 46)	49.03	6.67			

* Differences tested with unpaired *t*-test. VD, vascular density; SCP, superficial capillary plexus; DCP, deep capillary plexus.

**Table 3 biomedicines-13-01251-t003:** Multivariate linear regression model.

MODEL	F, *p* *	Gender		Age		Interaction Age × Group	
		Unstandardized B	*p* *	Standardized B	*p* *	Standardized B	*p* *
VD SCP—whole image (%)—initial	13.906; <0.001	2.85	0.003	−0.295	0.002		
VD SCP—whole image (%)—interaction	9.539; <0.001	2.643	0.007	−0.207	0.13	−0.128	0.359
VD SCP—fovea (%)—initial	0.415; 0.661	−0.905	0.582	−0.093	0.391		
VD SCP—fovea (%)—interaction	0.328; 0.805	−0.746	0.661	−0.136	0.377	0.063	0.690
VD SCP parafovea (%)—initial	7.895; <0.001	2.95	0.03	−0.246	0.016		
VD SCP parafovea (%)—interaction	5.977; <0.001	2.496	0.072	−0.104	0.464	−0.205	0.163
VD SCP perifovea (%)—initial	16.147; <0.001	3.103	<0.001	−0.205	0.002		
VD SCP perifovea (%)—interaction	10.914; <0.001	2.936	0.002	−0.233	0.083	−0.105	0.445
VD DCP whole image (%)—initial	8.860; <0.001	2.210	0.086	−0.307	0.003		
VD DCP whole image (%)—interaction	5.845; <0.001	2.229	0.095	−0.313	0.03	0.009	0.953
VD DCP fovea (%)- initial	1.323; 0.271	0.720	0.704	−0.148	0.172		
VD DCP fovea (%)—interaction	0.874; 0.458	0.694	0.724	−0.142	0.355	−0.009	0.956
VD DCP parafovea (%)—initial	8.767; <0.001	1.808	0.081	−0.303	0.003		
VD DCP parafovea (%)—interaction	6.393; <0.001	1.49	0.16	−0.179	0.207	−0.18	0.217
VD DCP perifovea (%)—initial	6.743; 0.002	1.927	0.183	−0.288	0.006		
VD DCP perifovea (%)—interaction	4.451; 0.006	1.693	0.190	−0.298	0.042	0.015	0.921

* Multiple linear regression analysis. VD, vascular density; SCP, superficial capillary plexus; DCP, deep capillary plexus.

**Table 4 biomedicines-13-01251-t004:** Hierarchical linear regression predicting vessel density (VD) in diabetic eyes.

Model		Unstandardized B	Standard Error	Standardized B	*p*	95% CI Lower	95% CI Upper
M₁	age	−0.095	0.058	−0.226	0.108	−0.212	0.022
	gender	1.719	1.240		0.172	−0.775	4.213
M₂	age	−0.090	0.057	−0.213	0.118	−0.204	0.024
	gender	0.979	1.260		0.441	−1.556	3.514
	HBA1c value	−1.223	0.615	−0.278	0.053	−2.460	0.015
M₃	age	−0.132	0.055	−0.313	0.020	−0.243	−0.022
	HBA1c	−1.048	0.578	−0.238	0.076	−2.211	0.115
	duration	−0.022	0.008	−0.387	0.007	−0.038	−0.006
	gender	2.151	1.248		0.092	−0.362	4.664
Model	R2	df1		df2		*p*	
M₁	0.096	2		48		0.089	
M₂	0.166	1		47		0.053	
M₃	0.288	1		46		0.007	

**Table 5 biomedicines-13-01251-t005:** Descriptive data for ganglion cell complex (GCC) parameters.

Parameter	Group	Arithmetic Mean	Standard Deviation	Difference	t	*p* *
GCC total	Diabetic patients	98.55	8.53	−1.88	−0.979	0.298
	Control	100.43	8.31			
GCC superior	Diabetic patients	97.94	9.26	−1.55	−0.756	0.452
	Control	99.50	8.80			
GCC inferior	Diabetic patients	99.17	8.31	−2.22	−1.184	0.240
	Control	101.38	8.10			
GCC FLV	Diabetic patients	1.06	1.17	0.13	0.334	0.051
	Control	0.93	2.05			
GCC GLV	Diabetic patients	2.84	3.76	1.04	1.404	0.164
	Control	1.80	2.79			

* Differences tested with unpaired *t*-test. GCC, ganglion cell complex; FLV, focal loss volume; GCL, global loss volume.

**Table 6 biomedicines-13-01251-t006:** Spearman test of correlation between duration of diabetes and three parameters.

Parameter	Duration of Diabetes
Non-Flow FAZ (mm^2^)	*ρ* = 0.249 (*p* = 0.289)
Vascular Density—Whole Image	*ρ* = −0.134 (*p* = 0.573)
GCC TOTAL	*ρ* = −0.001 (*p* = 0.970)

## Data Availability

The original contributions presented in this study are included in the article.

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
