# Peer review of "HD-OCT Angiography and SD-OCT in Patients with Mild or No Clinically Apparent Diabetic Retinopathy"

_biomedicines, 2025, doi:10.3390/biomedicines13051251_

Round 1
Reviewer 1 Report
Comments and Suggestions for Authors
This study applied High-Definition Optical Coherence Tomography Angiography (HD-OCTA) and Spectral-Domain Optical Coherence Tomography (SD-OCT) to quantitatively assess microvascular and structural changes in the retina and choroid. The study compared healthy controls to patients with no or early diabetic retinopathy. Measurements of vascular density (VD), foveal avascular zone (FAZ) area, retinal thickness, and ganglion cell complex (GCC) parameters were obtained. The authors emphasize the potential of HD-OCTA and SD-OCT as objective biomarkers for monitoring disease progression.
I believe this work has merit and can be recommended for publication, however, the following suggestions and comments may help improve the manuscript:
-
It would be interesting to discuss whether the observed results differed significantly between male and female subjects. Were there statistically significant or clinically meaningful differences in any of the measured parameters based on sex? Reporting on such analyses would strengthen the study.
-
Multivariate Regression with Age/Sex Correction. Investigating how correction for potential confounding co-factors, particularly age and sex, in a multiple regression model would alter the results presented in Tables 2 and 3 could further refine the analysis. Re-analyzing the data with these adjustments would allow for a more robust assessment of the independent effects of diabetic retinopathy on the measured retinal parameters.
-
The authors are encouraged to include a dedicated study limitations section. The retrospective nature of the study, relying on previously collected data, inherently prevents drawing conclusions about causality. The cross-sectional design further restricts findings to a single point in time, limiting the objective investigation of temporal sequences and cause-and-effect relationships during disease progression (e.g., whether vascular changes precede GCC changes, or vice versa). The study also employs a relatively modest sample size (54 eyes from diabetic patients and 58 eyes from controls), which may impact the power to detect subtle but clinically relevant differences. Furthermore, the authors have already acknowledged a potential selection bias related to sex ratio, and any other potential sources of bias should be addressed here.
Author Response
Maja Vinković
University Hospital Osijek
majavinkovic77@gmail.com
8th May2025
Manuscript ID: biomedicines-3582781
Title: HD-OCT angiography and SD-OCT in patients with mild or no clinically apparent diabetic retinopathy
To the Editor and Reviewers of Biomedicines,
We sincerely thank the Editor and Reviewers for their thoughtful and constructive comments on our manuscript. We have carefully considered each suggestion and have revised the manuscript accordingly.
Below is a detailed, point-by-point response to the reviewers' comments. Changes made in the manuscript are indicated with tracked changes.
REVIEWER 1
Comment 1: It would be interesting to discuss whether the observed results differed significantly between male and female subjects. Were there statistically significant or clinically meaningful differences in any of the measured parameters based on sex? Reporting on such analyses would strengthen the study.
Response: We thank the reviewer for this suggestion. In response, we performed additional analyses to evaluate potential sex-related differences in the measured parameters. Specifically, we compared vascular density (VD) in whole image, fovea, parafoveal and perifoveal region values between male and female subjects within both the control and patient groups.
Comment 2: Multivariate Regression with Age/Sex Correction. Investigating how correction for potential confounding co-factors, particularly age and sex, in a multiple regression model would alter the results presented in Tables 2 and 3 could further refine the analysis. Re-analyzing the data with these adjustments would allow for a more robust assessment of the independent effects of diabetic retinopathy on the measured retinal parameters.
Response: We have reanalyzed the data using multivariate regression models that correct for potential confounding factors, particularly age and sex. The results of these adjusted analyses have been added to Tables 3 and 4. We have updated the discussion section to reflect these adjusted findings and to further clarify the impact of these corrections.
Comment 3: The authors are encouraged to include a dedicated study limitations section. The retrospective nature of the study, relying on previously collected data, inherently prevents drawing conclusions about causality. The cross-sectional design further restricts findings to a single point in time, limiting the objective investigation of temporal sequences and cause-and-effect relationships during disease progression (e.g., whether vascular changes precede GCC changes, or vice versa). The study also employs a relatively modest sample size (54 eyes from diabetic patients and 58 eyes from controls), which may impact the power to detect subtle but clinically relevant differences. Furthermore, the authors have already acknowledged a potential selection bias related to sex ratio, and any other potential sources of bias should be addressed here.
Response: We have added a discussion of the methodological limitations in the Discussion section.
Reviewer 2 Report
Comments and Suggestions for Authors
The manuscript aimed to analyze retinal and choriocapillaris changes in diabetic patients with mild or no clinically apparent diabetic retinopathy using high-definition optical coherence tomography angiography (HD-OCTA) and spectral-domain OCT (SD-OCT). Results showed that diabetic patients had significantly reduced vascular density (VD) in the whole image and parafovea compared to healthy individuals, as well as focal thinning of the ganglion cell complex (GCC), which suggested early neurodegeneration. Although the foveal avascular zone (FAZ) was larger in diabetic patients, the difference was not statistically significant. The significance of this paper lies in its demonstration that advanced imaging techniques, such as high-definition HD-OCTA and spectral-domain SD-OCT, can detect early retinal vascular and neurodegenerative changes in diabetic patients before clinically apparent diabetic retinopathy. By identifying reduced vascular density and focal ganglion cell complex thinning, the study highlights the potential of these methods to serve as biomarkers for monitoring disease progression and visual prognosis. Furthermore, this is crucial for early intervention, as traditional diagnostic tools like ophthalmoscopy and fundus photography often fail to detect subtle, preclinical changes. Overall, the findings pave the way for improved management strategies to prevent DR progression and associated visual impairment.
- How reliable are the automated software algorithms used for measuring vascular density and retinal thickness? Could manual validation improve accuracy?
- Were there any biases in patient selection, such as the exclusion of patients with severe refractive errors or optical media opacities?
- The study found no significant differences in FAZ enlargement or GCC thinning between groups. Could this be due to limitations in sample size or imaging sensitivity?
Author Response
Maja Vinković
University Hospital Osijek
majavinkovic77@gmail.com
8th May2025
Manuscript ID: biomedicines-3582781
Title: HD-OCT angiography and SD-OCT in patients with mild or no clinically apparent diabetic retinopathy
To the Editor and Reviewers of Biomedicines,
We sincerely thank the Editor and Reviewers for their thoughtful and constructive comments on our manuscript. We have carefully considered each suggestion and have revised the manuscript accordingly.
Below is a detailed, point-by-point response to the reviewers' comments. Changes made in the manuscript are indicated with tracked changes.
REVIEWER 2
Comment 1: How reliable are the automated software algorithms used for measuring vascular density and retinal thickness? Could manual validation improve accuracy?
Response: We thank the reviewer for raising this important point. The automated segmentation and quantification algorithms employed in our study are embedded in the HD-OCTA and SD-OCT systems and have been validated in previous literature for their reproducibility and accuracy. To ensure reliability, we visually inspected all scans for segmentation errors or artifacts. Any scans with poor quality were excluded from analysis. For this study, we relied on high-quality automated results to ensure standardization and minimize bias.
Comment 2: Were there any biases in patient selection, such as the exclusion of patients with severe refractive errors or optical media opacities?
Response: To ensure optimal image quality and accurate segmentation, we excluded patients with high refractive errors (spherical equivalent > ±6.00 D), significant media opacities (such as dense cataract), and other retinal or optic nerve pathologies. These criteria were necessary to reduce imaging artifacts and ensure reliable quantitative analysis. We have reported exclusion critearia in Patients and methodes section.
Comment 3: The study found no significant differences in FAZ enlargement or GCC thinning between groups. Could this be due to limitations in sample size or imaging sensitivity?
Response: This finding could be due to limitations in sample size and imagining sensitivity; we have addresed it in limitations section.